# Maternal biomarkers of endothelial dysfunction and pregnancy outcomes in women with and without HIV in Botswana

Gaerolwe Masheto[1,2]*, Sikhulile Moyo[1,2,3], Terence Mohammed[1], Christine Banda[1], Charlene Raphaka[1], Gloria Mayondi[1], Joseph Makhema[1,2], Roger Shapiro[1,2,4], Mosepele Mosepele[1,2,5], Rebecca Zash[1,6,7], Shahin Lockman[1,2,6,7]

**1** Botswana Harvard AIDS Institute Partnership, Gaborone, Botswana, **2** Department of Immunology and Infectious Diseases, Harvard T.H. Chan School of Public Health, Boston, MA, United States of America, **3** Faculty of Health Sciences, School of Allied Health Professions, University of Botswana, Gaborone, Botswana, **4** Beth Israel Deaconess Medical Center, Boston, MA, United States of America, **5** Faculty of Medicine, University of Botswana, Gaborone, Botswana, **6** Brigham and Women's Hospital, Boston, MA, United States of America, **7** Harvard Medical School, Boston, MA, United States of America

* gmasheto@hsph.harvard.edu

**Data Availability Statement:** All relevant data are within the paper and its Supporting Information files.

## Abstract

### Background

Women living with HIV-1 (WLHIV) are at higher risk of having an adverse birth outcome, but the underlying mechanism(s) are unknown. We hypothesized that HIV-associated endothelial activation could adversely impact placental function and lead to impaired fetal growth or stillbirth.

### Methods

We used stored samples from WLHIV and HIV-negative women who had enrolled during pregnancy in the observational Botswana *Tshipidi* cohort. Written informed consent was obtained from the participants. We measured plasma levels of markers of endothelial activation (soluble vascular adhesion molecule 1 [VCAM-1], intercellular adhesion molecule 1 [ICAM-1] and E-selectin) from samples taken during pregnancy. We compared $log_{10}$ biomarker levels by maternal HIV status and by the timing of ART initiation (ART prior to conception vs. during pregnancy; ART prior to sample collection vs. no ART prior to sampling) using t-tests and the Kruskal-Wallis rank test. We evaluated the association between these biomarkers and adverse birth outcomes (composite of stillbirth or small for gestational age [SGA]) using univariate and multivariate log-binomial regression controlling for maternal age (continuous) and timing of ART start. We also used generalized linear models (GLM) to evaluate the association between continuous birthweight (in grams) and gestational age (in weeks) and markers of endothelial dysfunction, adjusting for maternal age (continuous) and timing of ART relative to sample collection.

### Results

Specimens collected before delivery were available for 414 women (372 WLHIV and 42 HIV-negative women), with a median age of 28 years and median gestational age at sample

**Funding:** Dr. Gaerolwe Masheto was supported by the Harvard University Center for AIDS Research (CFAR), an NIH-funded program (P30 AI060354), which is supported by the following NIH Co-Funding and Participating Institutes and Centers: National Institute of Allergy and Infectious Diseases, National Cancer Institute, Eunice Kennedy Shriver National Institute of Child Health and Human Development, National Heart, Lung, and Blood Institute, National Institute on Drug Abuse, National Institute of Mental Health, National Institute on Aging, National Institute of Diabetes and Digestive and Kidney Diseases, National Institute of General Medical Sciences, National Institute on Minority Health and Health Disparities, National Institute of Dental and Craniofacial Research, Office of AIDS Research, and Fogarty International Center. Dr. Sikhulile Moyo was partially supported through the Sub-Saharan African Network for TB/HIV Research Excellence (SANTHE 2.0), by Bill and Melinda Gates Foundation (INV-033558) and the National Institutes of Health NIH Fogarty International Center K43 TW012350-01. Dr. Shahin Lockman was supported by the National Institutes of Health NIH/ National Institute of Allergy and Infectious Diseases K24 mentoring grant - NIH K24 AI131928.

**Competing interests:** The authors have declared that no compering interest exist.

collection of 30 weeks (range 26, 35 weeks). WLHIV had significantly higher median VCAM1 (p = 0.002) than HIV-negative women, but HIV-negative women had higher median ICAM1 (p = 0.01); e-Selectin levels did not differ by maternal HIV status. Women starting ART during pregnancy had higher $\log_{10}$ VCAM1 levels than those on ART before conception, regardless of whether the sample was collected before (p = 0.02) or after (p = 0.03) ART initiation. However, ICAM1 and e-Selectin did not differ significantly by ART status or ART timing. Ninety-eight women (91 WLHIV and 7 HIV-negative), or 9 (2%) and 89 (22%) included in this study, had a stillborn or SGA baby respectively. Univariate and adjusted analyses did not show significant associations between levels of any of the biomarkers with these adverse birth outcomes. However, lower birthweight (p = 0.03) and lower gestational age at delivery were associated e-Selectin and ICAM (p = 0.008), respectively.

## Conclusion

Maternal HIV infection and lack of ART (or recent ART initiation) were associated with one marker of greater endothelial activation (VCAM-1), but not with other markers (ICAM-1 nor E-selectin) in pregnancy. e-Selectin was associated with lower birthweight and every unit increase in log ICAM-1 at delivery was associated with lower gestation age at delivery.

## Introduction

Increased access to 3-drug antiretroviral therapy (ART) during pregnancy has dramatically reduced new pediatric HIV infections globally and has improved maternal health [1–4]. However, women living with HIV (WLHIV) who take ART are at significantly higher risk of having an adverse pregnancy outcome (stillbirth, preterm delivery, and small for gestational age [SGA]) than women without HIV, particularly if ART is started prior to conception [5–14]. The underlying mechanisms for this increased risk are unknown. Given that more than 1.3 million WLHIV deliver annually [11] it is, therefore, crucial that we study the mechanism(s) underlying adverse birth outcomes in WLHIV, to help inform interventions to improve maternal and child outcomes.

Positive HIV status and ART use have both been independently associated with manifestations of endothelial dysfunction, including cardiovascular disease (CVD) [15–21]. Endothelial dysfunction is a term that covers diminished production/availability of nitric oxide and/or an imbalance in the relative contribution of endothelium-derived relaxing and contracting factors [22]. In the general population, high levels of endothelial adhesion molecules such as vascular cell adhesion molecule 1 (VCAM-1), intracellular adhesion molecule 1 (ICAM-1) and selectins (P-selectin or E-selectin) have been consistently associated with endothelial dysfunction [23–26]. These molecules work in concert to promote the development of atherosclerotic disease, with selectins facilitating activated leukocyte rolling and the adhesion molecules permitting adhesion and passage of leukocytes into the sub-endothelial space [26,27]. During pregnancy, endothelial function is critical for placentation, maternal volume expansion, and placental perfusion. Markers of endothelial dysfunction are associated with adverse birth outcomes in pregnant women without HIV [28], but similar data for pregnant women with HIV do not exist. We hypothesized that vascular endothelial dysfunction due to maternal HIV infection (and/or due to ART taken in pregnancy) adversely impacts placental perfusion during pregnancy, leading to an increased risk of placenta-mediated pregnancy complications such as fetal growth restriction (small for gestational age) and stillbirth. We, therefore, assessed

the relationship between (1) HIV, ART and the biomarkers VCAM-1, ICAM-1 and e-selectin, and (2) these biomarkers and adverse birth outcomes.

## Methods

### Study population

Our study utilized stored plasma samples and existing data collected in pregnant women who previously participated in the "Tshipidi" study in Botswana [29]. The Tshipidi study enrolled WLHIV and women without HIV ($\geq$18 years of age) during pregnancy or shortly after delivery from 2010 to 2012 and followed mothers and babies for 2 years postpartum to evaluate pediatric neurodevelopmental outcomes. The majority (88%) of women were enrolled before delivery at a median gestational age of 27 weeks. Maternal and infant blood samples were collected at enrollment, delivery, and 1, 6 and 24 months postpartum. Demographic information, HIV history (including the date of diagnosis, antiretroviral [ARV] regimen and date of ARV initiation, and most recent CD4 count and viral load), general medical history, and pregnancy outcome were recorded for all participants. Gestational age was estimated by the last menstrual period (which was ascertained using a combination of maternal self-report and written obstetric records that are completed for all pregnant women during antenatal care and labor/delivery, which nearly universally occur during admission to maternity wards). Written obstetric records also served as the source of birthweight and stillbirth data. Small for gestational age was calculated using estimated gestational age and birthweight.

Women received antiretroviral regimens according to Botswana national guidelines at the time. From 2008 to 2012, pregnant women with baseline CD4 count >250 cells/mm$^3$ and no WHO Clinical Stage 3 or 4 HIV illness received zidovudine. Women with baseline CD4 <250 cells/mm$^3$ or with WHO Clinical Stage 3 or 4 received 3-drug ART (generally zidovudine + lamivudine + nevirapine); the CD4 threshold was changed to <350 cells/mm$^3$ in 2012. Multiple gestations were excluded from this analysis.

The Tshipidi study (from which the de-identified data for this analysis came from) was approved by the Botswana Health Research Development Committee and the Office of Human Research Administration at the Harvard T.H. Chan School of Public Health. Mothers provided written informed consent for study participation. The work summarized in this paper was consistent with original study aims, and was also approved by IRBs as an amendment.

### Laboratory testing

One of the secondary objectives of the Tshipidi study was to investigate potential immune-mediated mechanisms that might explain adverse birth outcomes in HIV-exposed children. We evaluated this objective using stored maternal plasma samples from all participants who enrolled prior to delivery and who had known infant birth weight, gender, and gestational age. More samples were available for WLHIV than women without HIV at the time we conducted sample testing for this study because most plasma samples for women without HIV had been used by other studies. We measured plasma concentrations of endothelial dysfunction biomarkers ICAM-1 (detection range 0.3–20 ng/mL and sensitivity 0.057) ng/mL, VCAM-1 (detection range 6.3–200 ng/mL and sensitivity 1.26 ng/mL) and E-selectin (0.1–8 ng/mL and sensitivity 0.027 ng/mL) using commercial ELISA kits (R&D Inc, Minneapolis, USA), following the manufacturer's instructions [30]. Testing was done at the ISO 15189 accredited Botswana Harvard HIV reference laboratory.

## Statistical analysis

Biomarkers were analyzed as continuous variables. Timing of ART relative to sample collection was defined as 1) initiated pre-conception, 2) initiated post-conception, did not start ART by the time of sampling and, 3) initiated during pregnancy sample collection after ART start.

We compared maternal baseline characteristics using the two-sample t-test for continuous variables and Fisher's exact test for categorical variables. Log biomarker levels were compared by maternal HIV status and by the timing of ART initiation and sample collection using t-tests and the Kruskal-Wallis rank test. We also evaluated the association between these biomarkers and the adverse birth outcomes that are more likely to be related to endothelial dysfunction, namely stillbirth or small for gestational age (defined as $<10^{th}$ percentile weight-for-gestational age according to Intergrowth-21 standards) [31,32] using univariate and multivariate log-binomial regression controlling for maternal age (continuous) and timing of ART initiation. We also used the generalized linear models (GLM) to evaluate the association between continuous birthweight and gestational age and markers of endothelial dysfunction, adjusting for maternal age (continuous) and timing of ART relative to sample collection.

## Results

Among 454 women living with HIV and 458 women without HIV who were enrolled in Tshipidi, this analysis includes 372 women with HIV (82%) and 42 women without HIV (9%) who were enrolled in antepartum and had birth outcomes data and stored plasma collected before delivery available. Among women without HIV: women who had samples available for our analysis had slightly lower median gestation age [(39.5 vs 41 weeks; p = 0.001) but were similar in median age (p = 0.50), educational status (p = 0.90) and the number of stillbirths (p = 0.46) compared with HIV-negative women who did not have samples available. Among 372 WLHIV in our study sample, 346 (93%) had a known time of ARV/ART initiation of whom 39 started ART prior to conception, 181 started during pregnancy prior to enrollment (29 on ART and 152 on ZDV monotherapy) and 126 started after study enrollment/sample collection. Among the 26 women with unknown timing of ART start, 3 were on 3-drug ART, 3 on ZDV, 4 ARV naïve and 16 had no records. Table 1 below shows demographics and clinical characteristics of pregnant WLHIV and pregnant women without HIV. Women living with HIV were older, less educated and had lower income.

## Markers of endothelial dysfunction by maternal HIV status and timing of ART initiation

Fig 1 shows the levels of VCAM-1, ICAM-1 and e-selectin by maternal HIV status. WLHIV had a statistically significantly higher mean VCAM1 (2.90 ng/mL; 95% Confidence Interval (CI):2.88–2.93) than women without HIV (2.77 ng/mL; 95%CI: 2.72–2.83), Fig 1A. Women without HIV had a statistically significantly higher mean ICAM1 (2.42 ng/mL; 95%CI:2.40–2.44) compared to WLHIV (2.50 ng/mL;95% CI: 2.44–2.55), Fig 1B. No difference was observed in e-selectin levels by maternal HIV status (Fig 1C).

Endothelial biomarker level by receipt and timing of ART initiation and sample collection in relation to pregnancy is shown in Fig 2 (initiated pre-conception, initiated post-conception, or did not start ART by the time of sampling). ICAM-1 and e-selectin levels were not significantly different by the timing of ART relative to sample collection (Fig 2B and 2C). However, women starting ART during pregnancy had statistically significantly higher mean $\log_{10}$ VCAM1 levels (2.92 ng/mL; 95%CI:2.88–2.95) than women who were on ART before

**Table 1. Demographics and clinical characteristics of pregnant WLHIV and pregnant women without HIV (N = 414).**

| Variables | All women | Women living with HIV | Women without HIV | p-value |
|---|---|---|---|---|
| | N = 414 | N = 372 | N = 42 | |
| **Age (years)** | | | | |
| Median (Q1, Q3) | 28 (23, 32) | 23 (21, 30) | 28 (24, 33) | <0.001 |
| **Highest Education level, n (%)** | | | | |
| Junior Secondary school or lower | 278 (67) | 262 (71) | 16 (38) | <0.001 |
| Senior Secondary school | 99 (24) | 81 (21) | 18 (43) | |
| Tertiary | 36 (9) | 28 (8) | 8 (19) | |
| **Income per month (in US $), n, (%)** | | | | |
| <20 | 215 (55%) | 192 (54) | 23 (60) | 0.045 |
| 20–50 | 30 (8%) | 30 (8) | 0 | |
| 51–100 | 76 (19%) | 72 (20) | 4 (11) | |
| > 100 | 73 (18%) | 62 (18) | 11 (29) | |
| **Delivery type, n, (%)** | | | | |
| Vaginal delivery | 376 (95) | 338 (95) | 38 (100) | 0.154 |
| Caesarean section | 18 (5) | 18 (5) | 0 | |
| **Gestational Age at the time of blood collection (weeks), median (Q1, Q3)** | 30 (26, 35) | 30 (26, 35) | 34 (26, 37) | 0.044 |
| **Gestational age at delivery (weeks), median (Q1, Q3)** | 40 (38, 41) | 39 (38, 41) | 41(39, 42) | <0.001 |
| **Stillbirth, n, (%)** | 9 (2) | 8 (2) | 1 (3) | 0.063 |
| **Preterm delivery, n, (%)** | 53 (13) | 51 (14) | 2 (5) | 0.087 |
| **Small for Gestational Age (SGA), n, (%)** | 89 (23) | 83 (23) | 6 (16) | 0.52 |

conception, in both the samples that were collected before (2.83 ng/mL; 95%CI: 2.77–2.88; p = 0.02) or after (2.91 ng/mL; 95%CI: 2.87–2.944;p = 0.03) ART initiation, Fig 2A.

## Markers of endothelial dysfunction and adverse birth outcomes

Univariate and adjusted analyses did not show significant associations between levels of any of these biomarkers and the adverse birth outcomes of interest (stillbirth or SGA). Overall, 9 (2%) pregnancies ended in stillbirth and 89 (22%) with infants born SGA. Median ICAM-1 was 292 ng/mL [$Q_1$,$Q_3$: 214, 346] among stillbirths and 267 ng/mL [$Q_1$,$Q_3$: 209, 347] among live births (p = 0.89), median VCAM-1 was 868 ng/mL [$Q_1$,$Q_3$: 706, 1132]] among stillbirths and 780 ng/mL [$Q_1$,$Q_3$: 565, 1048] among live births (p = 0.36), and median e-Selectin was

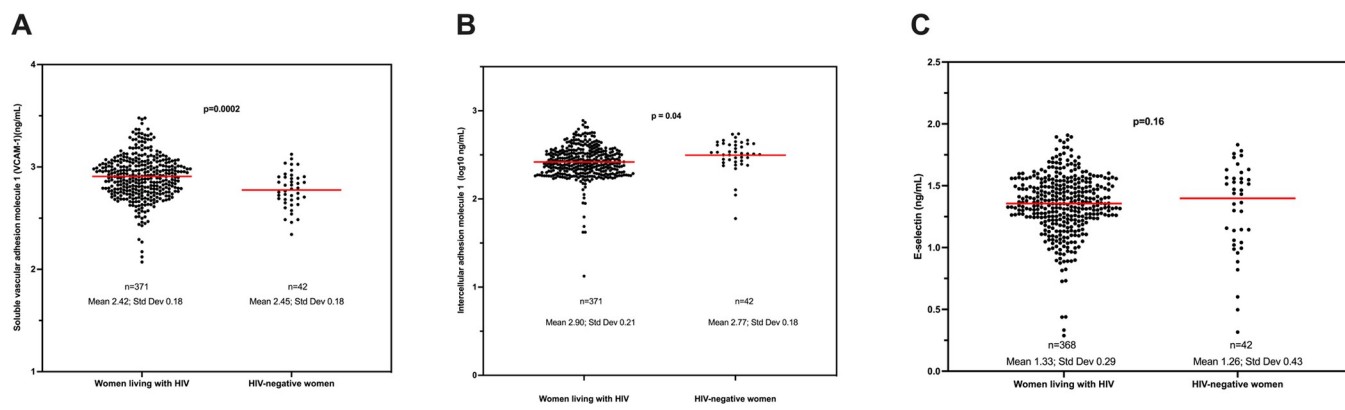

**Fig 1.**

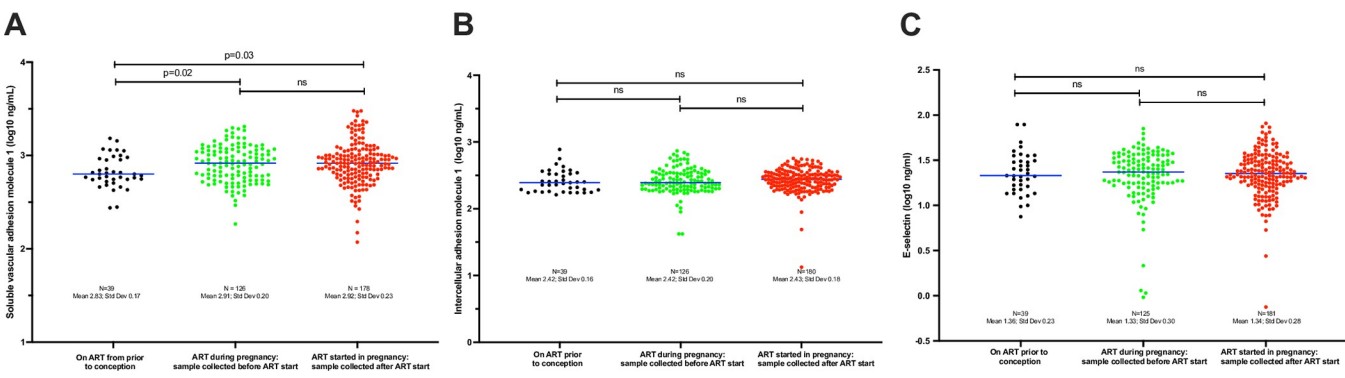

**Fig 2.**

24.2 ng/mL [$Q_1, Q_3$: 18.7, 27.3] among stillbirths and 22.7 ng/mL [$Q_1, Q_3$: 16.1, 33.3] among live births (p = 0.88). Median ICAM-1 was similar among SGA (276 ng/mL, $Q_1, Q_3$:199,349] and non-SGA (261 ng/mL, ($Q_1, Q_3$:199,349)]. Median VCAM was similar among SGA (740 ng/mL, $Q_1, Q_3$:569, 1104] and non-SGA (804 ng/mL, ($Q_1, Q_3$:586,1062)]. Median e-Selectin was also similar among SGA (24.6 ng/mL, $Q_1, Q_3$:17.5, 36.3] and non-SGA (22.5 ng/mL, ($Q_1, Q_3$:15.9,31.2)]. Every 1 unit increase in log e-Selectin (ng/mL) was associated with lower birthweight (β: -0.07, p = 0.03) and every 1 unit increase in log ICAM-1 (ng/mL) at delivery was associated with lower gestational age (weeks) at delivery (β: -0.06, p = 0.008), regardless of HIV status.

## Discussion

We evaluated whether maternal HIV status or timing of ART initiation was associated with biomarkers of vascular endothelial dysfunction during pregnancy and whether levels of these biomarkers were associated with SGA or stillbirth. We found that positive maternal HIV status was associated with significantly higher VCAM-1 levels, an indication of greater endothelial activation; however, positive maternal HIV status was not associated with increased levels of ICAM-1 or e-selectin. None of these three biomarkers were associated with the occurrence of stillbirth or SGA (although there we did observe a non-significant trend toward a higher risk of these adverse pregnancy outcomes with higher VCAM-1 levels, and we had limited power to evaluate stillbirth as an outcome).

Consistent with other studies for non-pregnant adults, median VCAM-1 levels were elevated in pregnant WLHIV compared with pregnant women without HIV [26–28], and there was no difference in e-selectin level by HIV status [26]. In the general population, elevated VCAM-1 has been consistently associated with endothelial dysfunction. Unlike prior data from non-pregnant adults [23,26], we found that median ICAM1 levels were higher in women without HIV compared with WLHIV. The reasons for this difference are not clear. ICAM-1 is an important molecule in immune-mediated and inflammatory processes and functions as a co-stimulatory signal which is important for the trans-endothelial migration of leukocytes and the activation of T cells [33,34]. Studies of ICAM-1 in women with HIV show no difference in the levels of ICAM-1 in normal pregnancy compared to non-pregnant women, but significantly increased levels of ICAM-1 in women with pregnancy-induced hypertension, especially preeclampsia [35,36].

We hypothesized that HIV-associated endothelial dysfunction would lead to a poorly functioning placenta and that there would be an increased prevalence of adverse birth outcomes

associated with abnormal placentas, including SGA and stillbirth. However, we found no association between these outcomes and biomarkers of endothelial dysfunction. This could indicate that the biomarkers we chose are not the most specific for endothelial dysfunction in pregnancy, or we may have lacked the power to detect differences given the relatively small study sample size.

Our study had several limitations. Our sample size was small for detecting associations with infrequent outcomes such as stillbirth. We could not measure all biomarkers of endothelial dysfunction and could not determine the causal relationship between endothelial dysfunction and adverse birth outcomes. However, our data suggest a possible role of e-selectin and ICAM-1 in lower birthweight and gestation age at delivery, respectively. This possible role of e-selectin and ICAM-1 in lower birthweight and gestation age at delivery is regardless of HIV status because there were no differences in these parameters by HIV status.

Not all stillbirths or SGA would result from placental dysfunction. Nevertheless, we believe that this study generated valuable pilot data in an area that has not been investigated thoroughly.

## Conclusion

Our study did not support endothelial activation as a mechanism for adverse birth outcomes in WLHIV, but further research is needed given the limitations and small sample size in this study.

## Supporting information

**S1 File.**
(XLS)

## Acknowledgments

We thank and acknowledge the Tshipidi Study participants and the investigators as well as the Botswana Harvard AIDS Institute Partnership leadership for their support.

## Author Contributions

**Conceptualization:** Gaerolwe Masheto, Mosepele Mosepele, Rebecca Zash, Shahin Lockman.

**Data curation:** Terence Mohammed, Christine Banda, Charlene Raphaka, Gloria Mayondi.

**Formal analysis:** Gaerolwe Masheto, Sikhulile Moyo, Joseph Makhema, Rebecca Zash, Shahin Lockman.

**Investigation:** Gaerolwe Masheto, Gloria Mayondi, Joseph Makhema, Rebecca Zash, Shahin Lockman.

**Methodology:** Terence Mohammed, Christine Banda, Charlene Raphaka, Mosepele Mosepele, Rebecca Zash, Shahin Lockman.

**Resources:** Sikhulile Moyo.

**Supervision:** Gaerolwe Masheto, Joseph Makhema, Roger Shapiro, Mosepele Mosepele, Rebecca Zash, Shahin Lockman.

**Validation:** Sikhulile Moyo.

**Writing – original draft:** Gaerolwe Masheto.

**Writing – review & editing:** Gaerolwe Masheto, Sikhulile Moyo, Joseph Makhema, Roger
Shapiro, Mosepele Mosepele, Rebecca Zash, Shahin Lockman.

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
