## [Decision Letter · Decision Letter 0]

18 Jul 2022

PONE-D-21-20063

Maternal biomarkers of endothelial dysfunction and pregnancy outcomes in women with and without HIV in Botswana

PLOS ONE

Dear Dr. Masheto,

Thank you for submitting your manuscript to PLOS ONE; I sincerely apologise for the unusually delayed review timeframe.

Your manuscript has been assessed by two reviewers, whose comments are appended below. After careful consideration, we feel that it has merit but does not fully meet PLOS ONE’s publication criteria as it currently stands. Although the reviewers find the topic of this study to be important, they raise concerns regarding whether elevated VCAM-1 in HIV-positive women was clinically or physiologically relevant, and the power of the study to detect significant differences between populations. Although the latter is discussed as a limitation, any revised manuscript should very clearly indicate the limitations of the current dataset and avoid drawing any conclusions (positive or negative) that cannot be fully supported by the analysis. We invite you to submit a revised version of the manuscript that addresses theses and other points raised during the review process.

We look forward to receiving your revised manuscript.

Kind regards,

Emily Chenette

Editor in Chief

PLOS ONE

Journal Requirements:

3. Thank you for stating in your Funding Statement: GM received award from the Harvard University Center for AIDS Research (CFAR), an NIH funded program (P30 AI060354), which is supported by the following NIH Co-Funding and Participating Institutes and Centers: National Institute of Allergy and Infectious Diseases, National Cancer Institute, Eunice Kennedy Shriver National Institute of Child Health and Human Development, National Heart, Lung, and Blood Institute, National Institute on Drug Abuse, National Institute of Mental Health, National Institute on Aging, National Institute of Diabetes and Digestive and Kidney Diseases, National Institute of General Medical Sciences, National Institute on Minority Health and Health Disparities, National Institute of Dental and Craniofacial Research, Office of AIDS Research, and Fogarty International Center. 

SM was partially supported through the Sub-Saharan African Network for TB/HIV Research Excellence (SANTHE), a DELTAS Africa Initiative [grant # DEL-15-006]. The DELTAS Africa Initiative is an independent funding scheme of the African Academy of Sciences (AAS)’s Alliance for Accelerating Excellence in Science in Africa (AESA) and supported by the New Partnership for Africa’s Development Planning and Coordinating Agency (NEPAD Agency) with funding from the Wellcome Trust [grant # 107752/Z/15/Z] and the UK government. 

SL was supported by the National Institutes of Health NIH/ National Institute of Allergy and Infectious Diseases K24 mentoring grant - NIH K24 AI131928. 

NO - All funders had no role in the study design, data collection and analysis, decision to publish, or preparation of the manuscript."

6. Please include a separate caption for each figure in your manuscript.

Reviewers' comments:

Reviewer's Responses to Questions

**Comments to the Author**

1. Is the manuscript technically sound, and do the data support the conclusions?

Reviewer #1: Partly

Reviewer #2: Partly

2. Has the statistical analysis been performed appropriately and rigorously? 

Reviewer #1: Yes

Reviewer #2: I Don't Know

3. Have the authors made all data underlying the findings in their manuscript fully available?

Reviewer #1: Yes

Reviewer #2: Yes

4. Is the manuscript presented in an intelligible fashion and written in standard English?

Reviewer #1: Yes

Reviewer #2: Yes

5. Review Comments to the Author

Reviewer #1: The authors have taken advantage of stored plasma samples and data collected from an observational study (Tshipidi) that enrolled in HIV-positive and HIV-negative pregnant women in Botswana 2010-2012 (a time when ART regimen in pregnancy was determined by CD4 count), following the participants for 2 years after enrollment, to try to assess whether elevation in three biomarkers of endothelial dysfunction may be in the causal pathway for increased risk of adverse pregnancy outcomes in HIV-infected women. This short report found only one marker of endothelial activation, VCAM-1, was elevated in HIV infection and with no/recent ART, and none of the markers were associated with birth outcome (stillbirth, SGA). Unfortunately, there were only limited samples remaining on the comparative HIV-negative women (9% vs 82% for HIV positive women), which could potentially restrict the power of the study to determine differences by HIV status.

Page 13, line 156: Table 1: Consider adding the “N” in the first row under the column titles so it is clear to reader.

Page 15, top paragraph lines 162-165, 170-172: Consider putting in the actual median values for the various markers (log ng/mL) as opposed to just listing the p values. Figure 1 on page 23 does not provide values, leaving the reader to guess what the median is in the comparative groups. A comment that while statistically significantly different by HIV status for VCAM-1 and ICAM-1 , visually the values do not look extremely different. For example, in figure 1 for VCAM-1, the median value looks like ~2.8 in HIV+ vs 2.6 log10 ng/mL in HIV- women. The clinical significance of such differences is not clear to me.

Discussion:

As noted above, the clinical significance of the differences observed is not clear. The authors note that “elevated VCAM-1” in the general population is associated with endothelial dysfunction. However, what levels define “elevated” that are associated with endothelial dysfunction is not discussed. Whether a decimal point difference in levels of the marker between groups (from visual inspection of the graphs) means that the marker is elevated enough to be associated with endothelial dysfunction is not clear. Additionally, references 23, 24 (referred to in line 202) report that while ICAM-1 was associated with risk of MI, levels of VCAM-1 were actually not, with VCAM-1 levels actually similar with and without MI (levels 638 vs 634 ng/mL). I did find a different reference suggesting a potential association (Wallen NH, Eur Heart J 1999, but p value for association was only 0.05).

Have these markers previously been associated with adverse pregnancy outcomes, or is this the first study that has looked at this association? There was only one study cited (ref 28 Chen X et al. PLosOne 2014) that did suggest an association of preterm delivery with elevated sCAM-1 and sVCAM-1 but that was not the adverse outcome being evaluated in this study. Are there associations with stillbirth and SGA?

Reviewer #2: This manuscript presents a prospective cohort study that evaluated differences in biomarkers of endothelial function in pregnancy by HIV and ART status as well as the relationship of the endothelial function biomarkers and birth outcomes. Overall, the research question is novel, and the study was well conducted; however, I think the manuscript could be improved in terms of the clarity of the reporting of the methods and the birth outcomes analyzed. My primary concern, which is also acknowledged as a limitation by the authors, is limited statistical power for the adverse birth outcomes outcomes examined.

1) Introduction – The introduction does a nice job laying out the current literature, but the research question is not entirely clear in the last sentence of the section. Based on the methods and results, I think the study had two components to the research question (1) assess the relationship of HIV, ART with the biomarkers and then (2) the relationship of the biomarkers with adverse birth outcomes. It seems part 1 – which takes up a significant proportion of results and discussion is currently missing from the research question.

2) Methods - Lab section –Were there limits of detection per the kits for each biomarker? If yes, please report and also how in the analysis how those beyond limits of detection contributed to the analysis.

3) Methods – it would be important for the methods for stillbirth determination, gestational age dating and birthweight data collection be defined. The exposure methods for biomarkers are clear but how outcome data was collected is not clear.

4) Preterm birth and Low birthweight –Why were preterm birth, birth weight, low birthweight and other commonly defined pregnancy outcomes not reported? There is certainly existing research and potential links between endothelial dysfunction and preterm birth.

5) Gestation duration, birthweight –Given the sample size, statistical power is a concern as noted by the study team. An analysis of continuous gestational age and birthweight which are commonly analyzed in pregnancy cohorts should be considered. Continuous birthweight-for-gestational age from Intergrowth, which is not as common as birthweight and gestation duration, could also be examined.

6) Stillbirth - There were 9 stillbirths in the cohort and as a result this cohort does not seem well positioned to evaluate the relationship between the biomarkers and stillbirth as a ‘primary’ outcome. While acknowledged in the discussion, it seems making no association with stillbirth a key message when there was almost no possibility of finding a difference is challenging and may be misinterpreted. This is in contrast to SGA, for which the sample size is not large, but there was certainly a chance of finding a moderate to large magnitude of effect if there was one.

Minor

1) Abstract – it would be helpful to report stillbirth and SGA rates separately to match analysis to be clear it was not a composite endpoint

2) Abstract and Discussion conclusion – The abstract conclusion focuses on difference by HIV and then the discussion conclusion focuses on birth outcomes. It seems both components are part of the research question and would be important to address in both sections.

3) Methods – the order of the analyses presented in the statistical analysis is confusing given the relationship of maternal factors with biomarkers is at the beginning and end of the section with the adverse birth outcomes in the middle.

4) Figure 1 – Please check labels – “ARV before pregnancy sampled before ARV start” – I believe this is women who started ARVs in pregnancy but the sample was collected before they started

5) Multiple gestation - where there any twins in the cohort examined? If so, how was this issue addressed in the statistical analysis.

6. PLOS authors have the option to publish the peer review history of their article (what does this mean?). If published, this will include your full peer review and any attached files.

Reviewer #1: No

Reviewer #2: No

---

## [Author Response · Author response to Decision Letter 0]

11 Oct 2022

Response to Reviewers Document attached.

---

## [Decision Letter · Decision Letter 1]

9 Nov 2022

PONE-D-21-20063R1Maternal biomarkers of endothelial dysfunction and pregnancy outcomes in women with and without HIV in BotswanaPLOS ONE

Dear Dr. Masheto,

Thank you for submitting your manuscript to PLOS ONE. After careful consideration, we feel that it has merit but does not fully meet PLOS ONE’s publication criteria as it currently stands. Therefore, we invite you to submit a revised version of the manuscript that addresses the points raised during the review process.

We look forward to receiving your revised manuscript.

Kind regards,

Ashish KC

Academic Editor

PLOS ONE

Journal Requirements:

Additional Editor Comments:

Dear Dr. Masheto

Thank you for the revision and proving point to point response to the reviewers' comment. There are some further recommendation made by the reviewers, we look forward to the updated manuscript with response to the reviewer's comment.

warm regards, ashish

Reviewers' comments:

Reviewer's Responses to Questions

**Comments to the Author**

1. If the authors have adequately addressed your comments raised in a previous round of review and you feel that this manuscript is now acceptable for publication, you may indicate that here to bypass the “Comments to the Author” section, enter your conflict of interest statement in the “Confidential to Editor” section, and submit your "Accept" recommendation.

Reviewer #1: All comments have been addressed

Reviewer #2: All comments have been addressed

2. Is the manuscript technically sound, and do the data support the conclusions?

Reviewer #1: Yes

Reviewer #2: Yes

3. Has the statistical analysis been performed appropriately and rigorously? 

Reviewer #1: Yes

Reviewer #2: Yes

4. Have the authors made all data underlying the findings in their manuscript fully available?

Reviewer #1: Yes

Reviewer #2: Yes

5. Is the manuscript presented in an intelligible fashion and written in standard English?

Reviewer #1: Yes

Reviewer #2: (No Response)

6. Review Comments to the Author

Reviewer #1: The reviewer comments have been generally addressed adequately.

The authors now newly note a possible role of e-Selection and ICAM in lower birthweight and gestation age at delivery at several points in the manuscript. It might be clearly noted that this was regardless of HIV status, given that there were no differences in these parameters by HIV status. So the relevance of these findings are really not HIV-specific but more relate to potential mechanisms of low birth weight and gestational age in general - think this might be made clearer as the paper is focused on differences due to HIV infection.

Reviewer #2: Thank you to the team for your thoughtful response to my comments.

I only have one small comment on the continuous analysis of birth weight and delivery. The Beta is given as -0.07 for example but as a reader I am not clear what the unit is in terms of weight same for the gestation age. As a result the magnitude of the association is not clear.

Is the birthweight kg or grams - is it 70 grams per log increase in log e-selectin? I assume for gestational age its weeks -0.06 weeks per log increase ICAM?

7. PLOS authors have the option to publish the peer review history of their article (what does this mean?). If published, this will include your full peer review and any attached files.

Reviewer #1: No

Reviewer #2: **Yes: **Christopher Sudfeld

---

## [Author Response · Author response to Decision Letter 1]

21 Dec 2022

The response to the reviewers comments has been uploaded.

---

## [Editor Report · Decision Letter 2]

5 Feb 2023

Maternal biomarkers of endothelial dysfunction and pregnancy outcomes in women with and without HIV in Botswana

PONE-D-21-20063R2

Dear Dr. Masheto,

We’re pleased to inform you that your manuscript has been judged scientifically suitable for publication and will be formally accepted for publication once it meets all outstanding technical requirements.

Kind regards,

Ashish KC

Academic Editor

PLOS ONE
---

## [Editor Report · Acceptance letter]

14 Feb 2023

PONE-D-21-20063R2 

Maternal biomarkers of endothelial dysfunction and pregnancy outcomes in women with and without HIV in Botswana 

Dear Dr. Masheto:

I'm pleased to inform you that your manuscript has been deemed suitable for publication in PLOS ONE. Congratulations! Your manuscript is now with our production department. 

Kind regards, 

on behalf of

Dr. Ashish KC 

Academic Editor

PLOS ONE